# Efficacy of duloxetine compared with opioid for postoperative pain control following total knee arthroplasty

**Man Soo Kim[1], In Jun Koh[2], Keun Young Choi[1], Sung Cheol Yang[1], Yong In[1]***

1 Department of Orthopaedic Surgery, Seoul St. Mary's Hospital, College of Medicine, The Catholic University of Korea, Seocho-gu, Seoul, Republic of Korea, 2 Department of Orthopaedic Surgery, Eunpyeong St. Mary's Hospital, College of Medicine, The Catholic University of Korea, Eunpyeong-gu, Seoul, Republic of Korea

* iy1000@catholic.ac.kr

**Data Availability Statement:** All relevant data are within the paper.

**Funding:** The authors received no specific funding for this work.

## Abstract

### Background

The purpose of this study was to assess the efficacy of duloxetine as an alternative to opioid treatment for postoperative pain management following total knee arthroplasty (TKA).

### Methods

Among 944 patients, 290 (30.7%) of patients received opioid or duloxetine for pain control for 6 weeks when the pain Visual Analogue Scale (VAS) score was greater than 4 out of 10 at the time of discharge. 121 patients in the Opioid group and 118 in the Duloxetine group were followed up for more than one year. Preoperative and postoperative patient reported outcome measures (pain VAS score, Western Ontario and McMaster Universities OA Index (WOMAC) score were compared. The rate of further drug prescription (opioid or duloxetine) after 6 weeks of first prescription, 30-day readmission rate, and side effects were also investigated.

### Results

There was no significant difference in pain VAS score, WOMAC Pain and Function score, at each time point between before and after surgery (all $p > 0.05$). Fifteen (9.8%) patients in the opioid group and six (4.4%) patients in the duloxetine group were prescribed additional medication after first 6 weeks, showing no significant ($p > 0.05$) difference in proportion. The 30-day readmission rate and the incidence of side effects were also similar (all $p > 0.05$). There was no difference in the incidence of side effects between the two groups ($p > 0.05$).

### Conclusion

Duloxetine and opioid did not show any difference in pain control, function, and side effects for up to one year after TKA. Although large-scale randomized controlled trials are still required to further confirm the side effects of duloxetine, it can be considered as an alternative to opioid for postoperative pain control following TKA.

**Competing interests:** The authors have declared
that no competing interests exist.

## Introduction

Postoperative pain in early days of recovery is severe after Total Knee Arthroplasty (TKA) [1].
Opioids have been used as an important part of multimodal postoperative analgesic regimen
[2, 3]. However, their use has negatively impacted not only patients, but also our society [4].
Recently, the use of excessive and indiscrete opioids has become a major concern [5, 6]. Continued use of opioid increases the risk of opioid-related side effect, opioid induced hyperalgesia, and addiction [2, 7, 8]. As a result, there has been growing interest in effective methods for
postoperative pain control while minimizing the use and side effects of opioids [3].

Selective serotonin and norepinephrine reuptake inhibitor (SNRI) can produce analgesic effect
on chronic pain of knee osteoarthritis (OA) [9]. In addition, SNRI is associated with increased
morphine's anti-nociceptive activity [10]. Duloxetine, a potent SNRI, can inhibit the reuptake of
serotonin and norepinephrine that could modulate descending inhibitory pain pathway in the
central nervous system (CNS) [11]. Therefore, theoretically duloxetine can reduce postoperative
pain after TKA [12, 13]. The role of this antidepressant on efficacy of postoperative pain control
has not been investigated in detail or limited to acute postoperative pain [12, 14].

Although limited data have suggested the efficacy of duloxetine in treating acute pain after
TKA, there is no study yet on the efficacy of duloxetine as a centrally acting agent compared to
opioids in postoperative pain control following TKA [15, 16]. In addition, its effects on pain
relief in short term and long term as part of a multimodal perioperative analgesic protocol for
TKA remain unknown [9, 12, 14]. Thus, the purpose of this study was to compare the efficacy
of duloxetine with that of opioid as part of a multimodal pain protocol including long-term
efficacy of pain relief following TKA.

## Methods

### Ethics statement

This study followed accepted ethical, scientific, and medical standards. It was conducted in
compliance with recognized international standards, including the principles of the Declaration
of Helsinki. This retrospective study was approved by the Institutional Review Board (IRB) of
Seoul St. Mary's Hospital, The Catholic University of Korea (IRB number: KC19RESI0574).

### General information

All patients who underwent unilateral primary TKA with posterior-stabilized (PS) prosthesis
by two surgeons in a single institution for the treatment of knee OA between January 2016 and
June 2018 with a minimum follow-up of one year were retrospectively reviewed. Patients were
eligible for inclusion in this study if they met the following criteria: the presence of moderate-
to-severe pain at discharge (postoperative one week) defined as an average movement pain
intensity score over 4 measured on a 0–10 Visual Analogue scale (VAS) (0, no pain; 10, the
worst possible pain) and requiring around-the-clock WHO step III opioids (strong opioids),
no previous treatment with opioids (i.e., opioid naïve) or duloxetine [17]. A self-reported questionnaire survey was conducted for patients at one week after surgery to determine the severity
of the pain directly recognized by the patient. This study was conducted for patients with a
VAS of 4 or more out of 10 points. Two surgeons used the same critical pathway and basic
pain management protocol for one week during hospitalization of the patients, but differed in
the addition of drugs for moderate to severe pain at discharge. One surgeon continued to use
opioids while another surgeon prescribed duloxetine for pain control. After discharge, an opioid and duloxetine were used for pain control at the same time point and were not converted
to other drugs based on a specific time point. Patients were excluded if they had a diagnosis of

osteonecrosis, inflammatory arthritis, traumatic OA, flexion contracture greater than 20˚, a previous infection, or knee surgery on the operated knee.

## Patients and treatment data

The number of knees that received primary TKA at our institution during the index period was 1,368 (1,181 patients). Initial candidates for analysis were 944 knees (944 patients) for unilateral TKA. Patient with moderate to severe pain was prescribed Oxycodone/Naloxone 10/5.0 mg (Targin 5/2.5 mg twice daily, Mundipharma co., Cambridge, UK) or duloxetine 30mg (Cymbalta 30mg per day, Lilly co., Indianapolis, IN, USA) for six weeks at the time of discharge (postoperative one week). Duloxetine 30mg was used without dose increment, referring to previous studies that proved sufficient effect even with duloxetine 30mg. Of 944 eligible patients with TKAs, 290 (30.7%) received an opioid or duloxetine for pain control postoperatively. There were 153 patients in the opioid group and 137 patients in the duloxetine group. Among patients who were treated with an opioid and duloxetine postoperatively, 121 patients in the opioid group and 118 patients in the duloxetine group were included in the final analysis (Fig 1).

## Operation and rehabilitation method

All operations were performed by two senior authors of this study under general anesthesia using posterior stabilized (PS) implants with tourniquet inflation to 300 mmHg. Both surgeons shared critical pathway except additional pain management regimen. Two hours before the surgery, multimodal oral analgesic drugs containing 200 mg celecoxib (Celebrex, Pfizer, NY, USA) once daily and 150 mg pregabalin (Lyrica, Pfizer, NY, USA) were administered as a preemptive analgesia. Neither surgeons performed periarticular injections or nerve blocks after operation. Instead, all patients received intravenous patient-controlled analgesia (PCA) which was programmed to deliver 1 mL of a 100-mL solution containing 2000 μg fentanyl postoperatively. Intravenous PCA was removed at four days after operation. Once patients restarted oral intake, 10 mg oxycodone, 200 mg celecoxib once daily, 37.5 mg tramadol (Paramacet, Donga, Seoul, Korea), and 650 mg acetaminophen (Tylenol, Janssen Korea, Seoul, Korea) were administered every 12 hours for seven days during hospitalization. Intramuscular injection of tridol (50 mg) was injected as an acute analgesic when a patient reported severe pain greater than level 6 on VAS of 0 to 10 during hospitalization period. After discharge, patients received 200 mg celecoxib once daily and 650 mg acetaminophen every 12 hours for six weeks after operation. In addition, patient with moderate to severe pain was prescribed Oxycodone/Naloxone or duloxetine for six weeks at the time of discharge. Opioid and duloxetine were continuously taken. Pain levels were evaluated at the outpatient clinic during follow-up at six weeks after discharge to determine whether additional duloxetine and opioid prescriptions were needed. Additional opioids and duloxetine were prescribed if the patient complained of sustained pain and required additional medication despite six weeks of pain medication after discharge. All patients were encouraged to perform active exercises according to our protocol of rehabilitation. Gradually increasing range-of-motion (ROM) and quadriceps-strengthening exercises were started immediately after the surgery. Patients began ambulation using a supportive device on the first postoperative day. Follow-up visits were scheduled at six weeks, three months, six months, and one year.

## Outcome measures

The primary outcome variable was self-reported pain severity with ambulation, which was assessed by using 10-point VAS score. Secondary outcome variables included pain at rest, pain at nighttime, and the Western Ontario and McMaster Universities Osteoarthritis Index

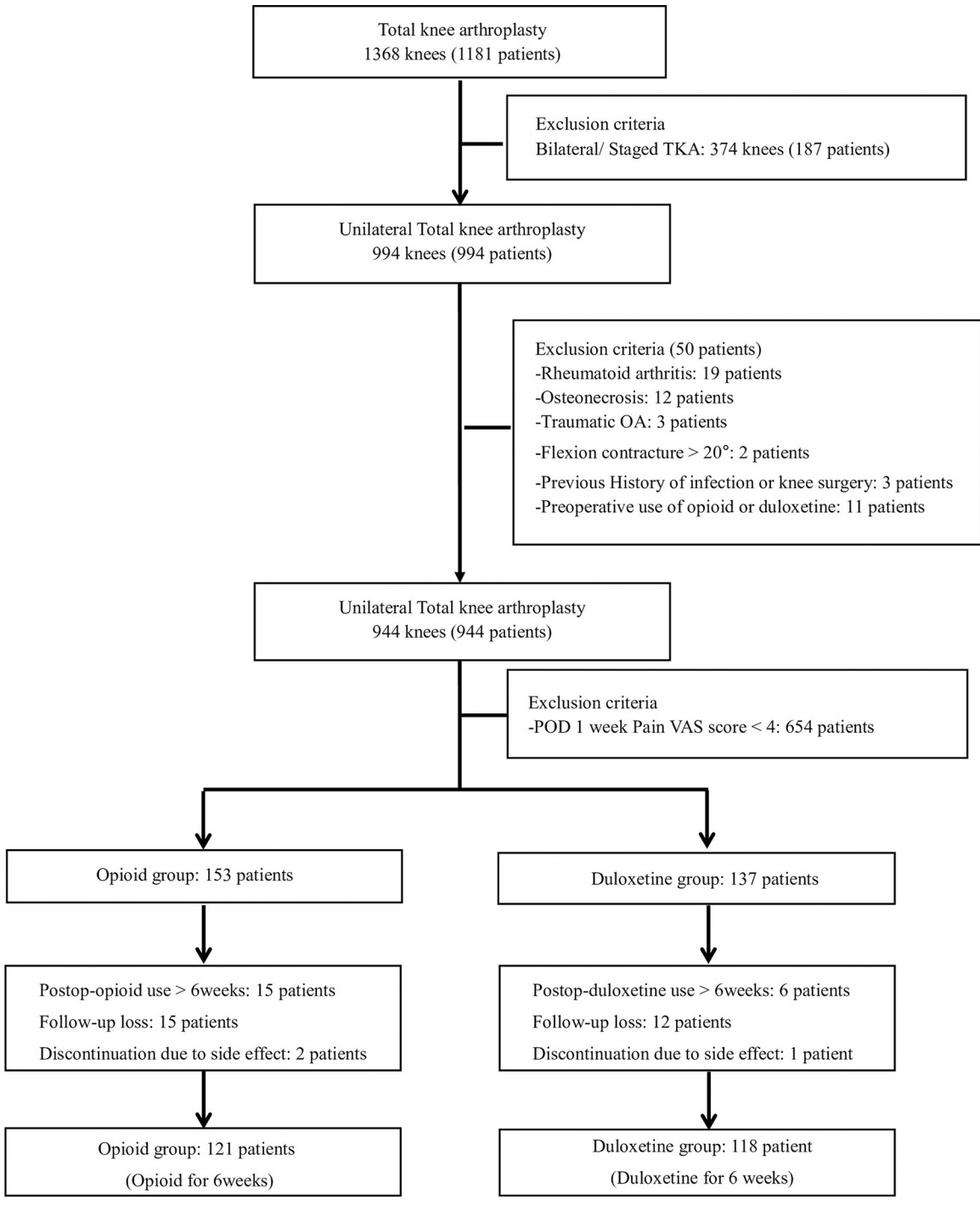

**Fig 1. Flow chart describing patients invited to participate in this study and included in the final analysis.**

(WOMAC) score. PCA consumption following TKA was investigated. The rate of further drug prescription (opioid or duloxetine) after six weeks of first prescription was evaluated. Thirty-day readmission rate was also identified with any reason between the two groups. Finally, adverse events were evaluated at six weeks postoperatively based on previous studies

regarding the safety of opioid and duloxetine [6, 9, 12, 15, 18]. Although duloxetine is approved for knee osteoarthritis, it can increase blood pressure in hypertension patients [19, 20] and glucose level in diabetes patients [21]. In addition, duloxetine can induce hyponatremia [22]. When it is combined with nonsteroidal anti-inflammatory drugs (NSAIDs), the risk of bleeding can increase [23]. When it is used with metoprolol, serious side effects can occur due to drug interactions [24]. Serotonin syndrome can also be induced [25]. These side effects mentioned above related to duloxetine were also investigated in the present study.

## Statistical analysis

Primary and secondary endpoints in the two groups were compared. All data are presented as mean and standard deviation. Comparison of continuous variables between the two groups was performed using unpaired Student's t-test. Fisher exact test was used for analysis of categorical variables. Propensity score matching and all statistical analyses were performed using SPSS ver. 21.0 program (SPSS Inc., Chicago, IL, USA). A $p$-value $< 0.05$ was considered statistically significant.

## Results

Fifteen (9.8%) patients in the opioid group and six (4.4%) patients in the duloxetine group were prescribed additional medication after first six weeks of pain control using opioid or duloxetine ($p = 0.111$) (Fig 1). In addition, none of the patients who took the drug for more than 6 weeks in either group took the drug for more than 3 months. Finally, there were 121 patients in the opioid group and 118 patients in the duloxetine group (Fig 1). Characteristic of these patients are shown in Table 1. Both groups had similar postoperative pain at postoperative six weeks,

Table 1. Patient demographics and preoperative characteristics*.

| | Opioid (n = 121) | Duloxetine (n = 118) | p-value |
|---|---|---|---|
| **Demographic data** | | | |
| Age (years) | 71.3 (7.2) | 70.0 (7.0) | 0.181 |
| Gender (female)† | 105 (86.8) | 98 (83.1) | 0.472 |
| BMI (kg/m²) | 26.0 (2.8) | 25.5 (3.4) | 0.200 |
| ASA status† | | | 0.101 |
| 1 | 22 (18.2) | 33 (28.0) | |
| 2 | 98 (81.0) | 82 (69.5) | |
| 3 | 1 (0.8) | 3 (2.5) | |
| Tourniquet time (minutes) | 44.8 (11.4) | 43.3 (10.8) | 0.311 |
| Specific comorbidities | | | |
| Hypertension | 76 (62.8) | 74 (62.7) | 0.987 |
| Diabetes | 22 (18.2) | 21 (17.8) | 0.938 |
| Cardiac disease | 15 (12.4) | 21 (17.8) | 0.243 |
| Cerebrovascular event | 5 (4.1) | 10 (8.5) | 0.191 |
| Thyroid disease | 11 (9.1) | 8 (6.8) | 0.509 |
| Kidney disease | 5 (4.1) | 5 (4.2) | 0.968 |
| Pulmonary disease | 8 (6.6) | 8 (6.7) | 0.586 |
| Liver disease | 7 (5.8) | 2 (1.7) | 0.097 |
| Depression | 1 (0.8) | 2 (1.7) | 0.619 |

* Data are presented as mean (standard deviation).

† Data are presented as number (percentage).

BMI = body mass index; ASA = American Society of Anesthesiologist

**Table 2. Pain VAS on walking, nighttime, and resting*.**

| | Walking | | | Nighttime | | | Resting | | |
|---|---|---|---|---|---|---|---|---|---|
| | Opioid (n = 121) | Duloxetine (n = 118) | p-value† | Opioid (n = 121) | Duloxetine (n = 118) | p-value† | Opioid (n = 121) | Duloxetine (n = 118) | p-value† |
| Preop | 7.2 (1.0) | 7.1 (0.8) | 0.162 | 5.2 (1.6) | 5.3 (1.6) | 0.421 | 3.8 (1.7) | 3.6 (1.6) | 0.347 |
| POD 6W | 3.5 (1.6) | 3.4 (1.5) | 0.430 | 3.5 (1.8) | 3.3 (1.6) | 0.551 | 1.8 (1.3) | 1.6 (1.2) | 0.207 |
| POD 3M | 2.8 (1.4) | 2.7 (1.5) | 0.720 | 2.3 (1.6) | 2.3 (1.7) | 0.955 | 1.3 (1.3) | 1.3 (1.5) | 0.866 |
| POD 6M | 2.2 (0.9) | 2.2 (0.8) | 0.245 | 1.5 (1.6) | 1.1 (1.4) | 0.080 | 1.1 (1.1) | 0.9 (0.9) | 0.115 |
| POD 1Y | 1.8 (1.2) | 1.9 (1.1) | 0.682 | 1.1 (1.1) | 0.9 (1.0) | 0.063 | 0.7 (0.8) | 0.6 (0.7) | 0.252 |

* Data are presented as mean (standard deviation).

POD = postoperative; W = week; M = month; Y = year

three months, six months, and one year during ambulation, at nighttime, and at rest (all $p>0.05$) (Table 2). ROM did not differ between the two groups preoperatively or postoperatively (all $p>0.05$) (Fig 2). PCA consumption was 52.6 ml in the opioid group and 55.3 ml in the duloxetine group without showing significant difference ($p = 0.492$). When the use of PCA was compared based on BMI 24 [2], it showed no significant difference between the group with BMI > 24 and the group with BMI ≤ 24 in either opioid or duloxetine users (all $p > 0.05$).

WOMAC pain score did not differ between the two groups either (all $p>0.05$). WOMAC Stiffness, Function, and Total score showed no significant difference between the two groups either at each time point (all $p>0.05$) (Fig 3).

The 30-day readmission rate was similar between the two groups (duloxetine group vs. opioid group: 3 (2.5%) vs. 1 (0.8%), $p>0.05$). Two patients in the duloxetine group and one patient in the opioid group were readmitted for pain control. The one patient in the duloxetine group was due to superficial wound infection.

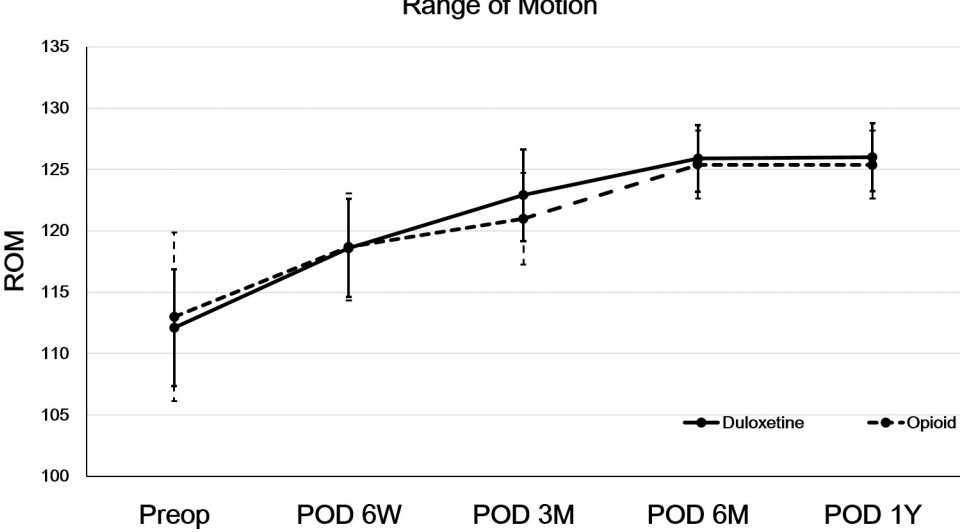

**Fig 2. Comparisons of Range of Motion (ROM) between duloxetine and opioid groups.** ROM did not differ between the two groups during postoperative period. Both groups showed significant improvement of ROM at 1 year postoperatively compared to the preoperative period ($p < 0.05$). Error bars represent the standard deviation.

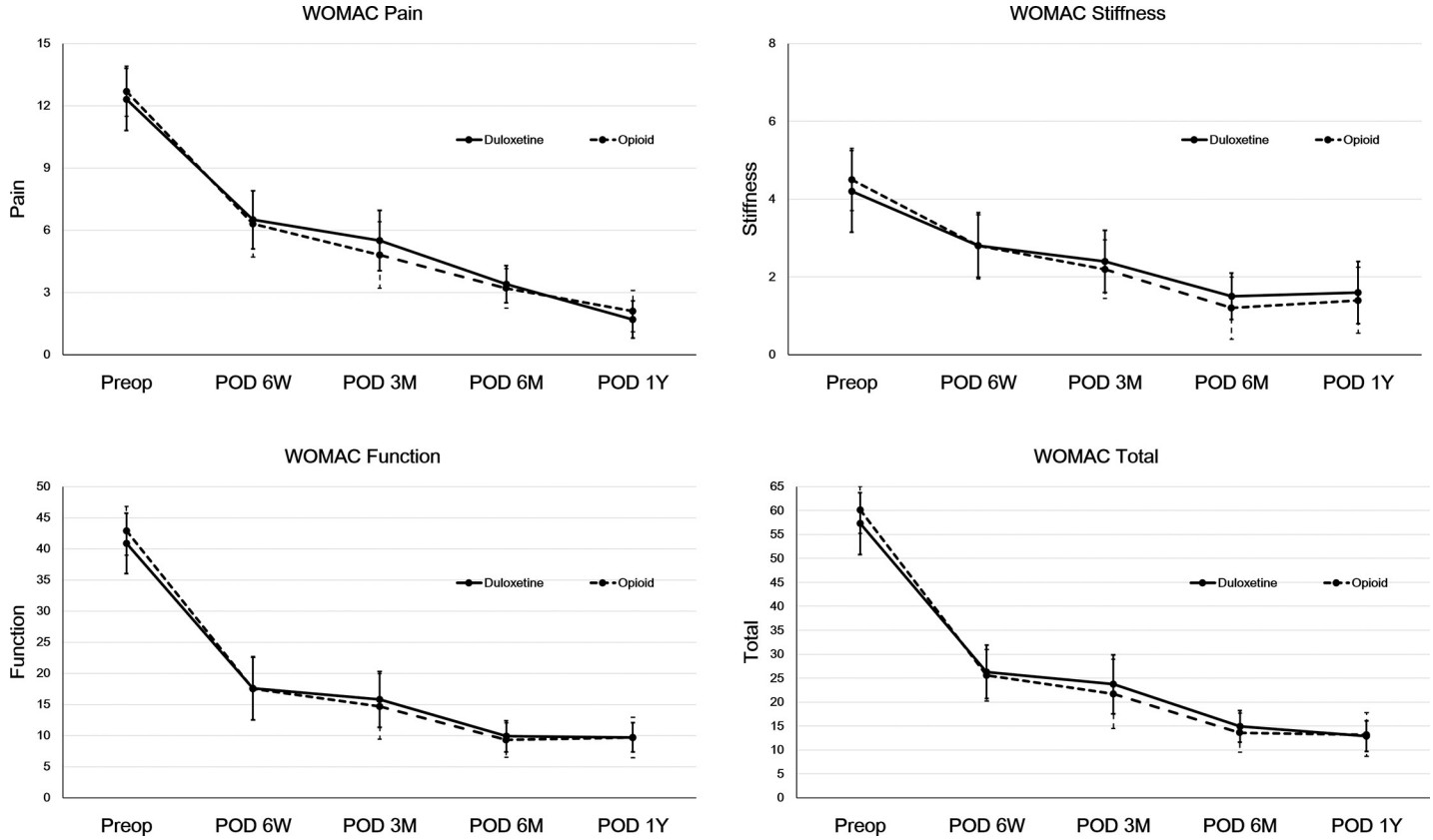

**Fig 3. Comparisons of Western Ontario and McMaster Universities OA Index (WOMAC) subscores between duloxetine and opioid groups.** There was no difference between the two groups regarding WOMAC pain (A), WOMAC stiffness (B), WOMAC function (C), or WOMAC total (D) at postoperative 6 weeks, 3 months, 6 months, or 1 year. Both groups showed significant improvement of WOMAC pain, stiffness, function, and total score at 1 year postoperatively compared to the preoperative period (all $p < 0.05$). Error bars represent the standard deviation.

There was no difference in the incidence of side effects between the two groups ($p > 0.05$) (Table 3). Two patients in the opioid group stopped taking the medication due to nausea, vomiting, and drowsiness. One patient in the duloxetine group stopped taking the medication due to drowsiness.

Hypertension patients and diabetes patients did not receive additional treatment or change treatment due to increased blood pressure or blood sugar by taking duloxetine. Bleeding-related symptoms and hyponatremia after taking duloxetine were not observed.

**Table 3. Incidences of adverse events\*.**

| Adverse events | Opioid (n = 121) | Duloxetine (n = 118) | *p*-value |
|---|---|---|---|
| Nausea/vomiting | 21 (17.4) | 14 (11.9) | 0.274 |
| Constipation | 16 (13.2) | 13 (11.0) | 0.693 |
| Dizziness | 21 (17.4) | 19 (16.1) | 0.863 |
| Drowsiness | 27 (22.3) | 22 (18.6) | 0.524 |
| Headache | 23 (19.0) | 18 (15.3) | 0.495 |
| Dry mouth | 27 (22.3) | 25 (21.2) | 0.876 |

\*Data are presented as number (percentage) of patients.

## Discussion

The most important finding of this study was that administration of duloxetine for six weeks had similar efficacy to that of opioid for pain and functional improvement following TKA. In addition, duloxetine did not cause major side effects when compared to opioid.

In general, the dosage of duloxetine starts with 30 mg initially. It is increased to 60 mg, because this drug might have side effects [26]. In a previous randomized controlled trial, postoperative pain and quality of life in patients who underwent TKA for OA were improved by 30 mg of duloxetine [27]. In addition, 30 mg of duloxetine showed sufficient effect on fibromyalgia [28]. Therefore, the dosage of duloxetine was chosen to be at 30 mg in the duloxetine cohort.

Historically, the use of opioids has been limited to acute pain management or diseases related to chronic pain [28]. However, as thresholds for the use of opioids have gradually decreased, overuse, misuse, and dependency of opioids have become major issues in recent years [5, 6]. These opioid epidemics are emphasizing the importance of an effective way to improve management of pain control after orthopedic procedure without dependency [29]. Despite the above data, little is known about the effective substitution of opioids for management of pain after TKA. Therefore, there have been a growing demand for methods to reduce or replace the use of opioids after TKA.

In this study, we found that postoperative use of duloxetine for six weeks in patients who had moderate pain level at discharge showed similar pain control to opioid after TKA during postoperative 1, 3, 6, and 12 months. Several positive attempts have provided improved expectations for the use of antidepressants for postoperative pain control [30]. YaDeau et al. [16] and Ho et al. [15] have studied perioperative use of duloxetine following TKA. They showed that duloxetine could reduce opioid consumption without affecting pain level or side effects [15, 16]. These findings were partly different from our study results. Duloxetine does not have an analgesic effect during immediate postoperative period of two weeks after TKAs [27]. Therefore, evaluating the efficacy of duloxetine with two weeks might not be an obvious result. Unlike those studies described above, we compared results for up to one year after TKA with the use of the two drugs for six weeks after surgery. This is the first study to identify short- and long-term course of two drugs over a period of time after TKA rather than temporary drug use [15, 16]. Our findings of this study suggest that incorporation of duloxetine into current after-discharge multimodal pain management protocol might be an effective alternative for patients who need to use opioids because of sustained post-discharge pain after TKA.

Major barriers to opioid use are opioid-related side effects [31]. The most common side effects are constipation, dizziness, nausea, vomiting, and delirium. Duloxetine is a drug that can act on the central nervous system like opioids. However, it has distinctly different mechanisms of action and other side effects [32]. The most common side effects of duloxetine are nausea, vomiting, constipation, and dry mouth [23, 33]. In the present study, side effects of duloxetine and opioids were found to be comparable with each other. The frequency of these side effects in the present study was similar to those reported in previous studies [9, 18, 34]. Nonetheless, the safety of duloxetine is inconclusive with supporting evidence in the present study.

Duloxetine might increase blood pressure after administration due to its cardiovascular side effects and its noradrenergic effect. [19, 20]. However, there was no restriction on the use of duloxetine in patients with high blood pressure as discontinuation alone could improve hypertension caused by the use of duloxetine [19, 20]. Duloxetine resulted in modest increases in fasting plasma glucose in short- and long-term studies [21]. The short-term study used duloxetine 60 mg for 12 weeks and the long-term study used drugs for up to 52 weeks [21]. In

this study, hypertension patients and diabetes patients did not receive additional treatment or change treatment due to increased blood pressure or blood sugar after taking duloxetine [19–21].

When metoprolol and antidepressant are used together, serious complications such as severe bradycardia and atrioventricular block can occur [24]. However, such risk is relatively low in the case of escitalopram, citalopram, and duloxetine [24]. Nevertheless, when used together, serotonin syndrome can occur due to drug interactions [25]. It is necessary to observe carefully at all times. In this study, preoperative drug identification was used to identify drugs that patients were taking. In the case of patients using metoprolol and diuretics, the use of duloxetine was restricted. No patients took metoprolol and duloxetine together. No patients developed serotonin syndrome either.

When duloxetine and NSAID are used together, the risk of bleeding increases due to platelet dysfunction [23]. Since duloxetine is frequently used for OA-related pain, for which NSAIDs are not effective, these two drugs are often used in combination. In this study, epistaxis, hematochezia, hematocrit reduction, and minor bleeding-related symptoms such as bruising were not observed in relation to the use of duloxetine. No major bleeding findings requiring medical intervention were confirmed.

In the case of selective Cox-2 inhibitor, metabolism is achieved in the liver by cytochrome P450 (CYP) enzymes. However, duloxetine can inhibit the activity of CYP enzymes. Thus, drug-drug interactions may occur when two drugs are used in combination [35, 36]. In this study, basically all patients were administered celecoxib, a selective cox-2 inhibitor. When duloxetine was used in combination with celecoxib in these patients, it was not necessary to extend the duration of drug use for all patients. Since only a few patients took the drug for more than 6 weeks, it did not appear that the duration of drug use was prolonged due to an effect on distribution or clearance by co-medication.

Since duloxetine is highly bound to plasma proteins and its metabolites are excreted by kidneys, caution should be exercised in cases of impaired renal function [37]. However, for patients with mild to moderate renal impairment, it is not necessary to adjust the dose of duloxetine. For patients with ESRD or severe renal impairment, duloxetine is generally not recommended [37]. Duloxetine was not administered in patients with more severe findings than mild to moderate renal impairment through a preoperative renal function test.

It is well known that the use of antidepressants can increase the risk of hyponatremia [22]. However, among antidepressants, duloxetine has a lower incidence rate ratio of hyponatremia compared to citalopram [22]. Although electrolyte tests were not continuously performed in this study, there were no cases of hyponatremia in the electrolyte test performed at two weeks after drug administration. Since the risk of hyponatremia is greatly increased when duloxetine and diuretic treatment are used at the same time, in this study, drugs being taken by patients were identified through preoperative drug identification. The use of duloxetine was restricted for patients using diuretics.

This study has certain limitations. First, nearly all (203/239, 85%) patients were women. The reason for such predominance of Korean female patients remains unknown, although such predominance has been well-documented [38–41]. Second, this study was a retrospective comparative study based on the database of one institution. It was difficult to control confounding factors with possible selection bias. Therefore, it is difficult to generalize results of this study to other populations. We attempted to minimize population heterogeneity by excluding patients who used opioids and duloxetine prior to surgery. We intend to conduct a randomized controlled trial comparing duloxetine and opioids prospectively for postoperative pain control after TKA in the future. Third, because the same multimodal pain regimen including opioid was used until five days postoperatively during hospitalization, there was no

result of early postoperative effect of pain control between the two groups. However, the purpose of this study was to see long-term pain control effects of 6-week administration of both drugs. Thus, we focused on efficacy of short- and long-term pain control in both groups. Fourth, since the number of patients in this study was insufficient, we could not judge side effects of medication. Further study with more patients is warranted. Fifth, since duloxetine is an antidepressant drug, it is important to evaluate the mood of patients. Although it would be nice to evaluate it using the Hamilton depression scale [27], we did not perform it in this study. Finally, results of this study could only be extrapolated to patients undergoing TKA. Other orthopedic procedures might require further studies. Despite these limitations of the methodology, we believe that results of the present study have great strengths in that it is the first study about agents that could replace opioids in patients with persistent pain after TKA.

## Conclusion

In conclusion, duloxetine and opioid did not show any difference in pain control, function, or side effects for up to one year after TKA. Although high quality large-scale randomized controlled trials are still required to further confirm the side effects of duloxetine, our results indicate that it is a good alternative to opioid in a multimodal pain management protocol for patients who are discharged after TKA.

## Author Contributions

**Conceptualization:** Man Soo Kim, Yong In.

**Data curation:** Man Soo Kim, Keun Young Choi, Sung Cheol Yang.

**Formal analysis:** Man Soo Kim, In Jun Koh, Sung Cheol Yang.

**Investigation:** Man Soo Kim, Keun Young Choi, Sung Cheol Yang.

**Methodology:** Man Soo Kim, Keun Young Choi.

**Resources:** Man Soo Kim.

**Software:** Man Soo Kim.

**Supervision:** In Jun Koh, Yong In.

**Validation:** Man Soo Kim, In Jun Koh, Yong In.

**Visualization:** Man Soo Kim, In Jun Koh, Yong In.

**Writing – original draft:** Man Soo Kim.

**Writing – review & editing:** Yong In.

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
