## [Decision Letter · Decision Letter 0]

18 Mar 2021

PONE-D-21-04527

Efficacy of Duloxetine Compared with Opioid for Postoperative Pain Control Following Total Knee Arthroplasty

PLOS ONE

Dear Dr. In,

Thank you for submitting your manuscript to PLOS ONE. After careful consideration, we feel that it has merit but does not fully meet PLOS ONE’s publication criteria as it currently stands. Therefore, we invite you to submit a revised version of the manuscript that addresses the points raised during the review process.

We look forward to receiving your revised manuscript.

Kind regards,

Vijayaprakash Suppiah, PhD

Academic Editor

PLOS ONE

Journal Requirements:

Reviewers' comments:

Reviewer's Responses to Questions

**Comments to the Author**

1. Is the manuscript technically sound, and do the data support the conclusions?

Reviewer #1: Partly

Reviewer #2: Yes

Reviewer #3: Partly

2. Has the statistical analysis been performed appropriately and rigorously? 

Reviewer #1: Yes

Reviewer #2: Yes

Reviewer #3: I Don't Know

3. Have the authors made all data underlying the findings in their manuscript fully available?

Reviewer #1: Yes

Reviewer #2: Yes

Reviewer #3: Yes

4. Is the manuscript presented in an intelligible fashion and written in standard English?

Reviewer #1: Yes

Reviewer #2: Yes

Reviewer #3: No

5. Review Comments to the Author

Reviewer #1: Yong In et al. present in their manuscript “Efficacy of Duloxetine Compared with Opioid for Postoperative Pain Control Following Total Knee Arthroplasty“ the alternative use of Duloxetin in contrast to opioid treatment in patients after total knee arthroplasty.

This retrospective study demonstrate no significant difference in pain control using Duloxetin instead of opioid treatment over 6 weeks after arthroplasty in patients with VAS pain score >4 after one week of operation.

The authors describe, that the Duloxetin medication was prescribed with a dose of 30mg. This dose is a very low dose, as for i.e. neuropathic pain the recommended dose is 60mg daily. Why the dose had been set to 30mg and not to 60mg? This aspect should be clarified and be considered in the discussion section of the manuscript.

Furthermore Duloxetin has a distinct profile of side affects which should be kept in mind.

In the manuscript this distinct side affects should be mentioned. However, this data are also mainly missing in the result section. 62.7% of the patients had a hypertension and 17.8% had a diabetes in the medical record. The question is in how many patients with diabetes had to treated additionally and in how many patients hypertension got worse.

Looking at the risk of medical interaction Duloxetin is not recommended for the use in addition to metoprolol, further it can induce a Serotonin syndrome. Are there any data about patients suffering from a Serotonin Syndrome in this patient cohort?

Mainly Duloxetin is increasing the risk of bleeding, especially in combination with NSAR medication. Are there any data in this study cohort? Further Hyponatriemia is a severe side affect in the treatment of patients with Duloxetin especially in patients with diuretic treatment. How was this risk managed in this patients.

The risk profile of Duloxetin should be discussed in comparison to the risk profile of opioid use in the indication of prolonged treatment after total knee arthroplasty. From my point of view there should be clear statement that the use of Duloxetin is still an off label use, which has to be kept in mind especially as the risk profile of these very good medication is not low.

Reviewer #2: Dear authors,

Here you receive my review concerning the manuscript entitled “Efficacy of Duloxetine compared with opioid for postoperative pain control following total knee arthroplasty” for PLOS one (PONE-D-21-04527). The authors from the orthopedic surgery department present the results from their retrospective non-randomized single center observational study. The study was approved by the local institutional review board with number provided.

This study number cannot be found in the database. Please explain? Line 82. Patients provided written informed consent. When were patients identified as eligible for this study and how were they approached for providing their informed consent. The reason for this question is in fact and rather to explain “retrospectively”.

Primary question of this study was to determine whether Duloxetine could substitute opioids in a multimodal pain relief protocol after discharge and identification and difference in side effects of the 2 analgesics.

Inclusion and exclusion criteria are well described. Language is clear and easy understandable.

Numbers of patients described in the “patients and treatment data” and abstract differ, i.e.857 patients vs 944 or 657 ? and 260 vs 290? I mean 153+137 is the mentioned 290, whereas the 121+118 is 239and not the 260 mentioned in the abstract. Moreover in the flow chart 944 differs from 857 in abstract or 944 in the patient and treatment data. Please rewrite with correct numbers?

So, patients received during the first postoperative period fentanyl PCA. Is it possible to differentiate between the needs of PCA fentanyl use during these first days in order to recognize possible differences in pain between patients. This may possibly have predictive value in need for analgesics after discharge. Maybe when possible correct these data for BMI differences between patients?

What was the definition for kidney disease? Did the presence of kidney disease not have an impact on prescription of either drug? How was drug safety for these patients guaranteed?

The patients , i.e. 15 in opioid group and 6 patients in the duloxetine group who received additional analgesics after six weeks. Did these patients also suffer from increased pain in the first postoperative week and needed more analgesic then also? Or did they have a (more) complicated course? Or tool these patients other comedication that may have had an effect on distribution or clearance, e.g. possible CYP enzyme competitors?

Is it possible to recognize difference between patients regarding pain sensation/severity after the first week?. In other words is it possible to recognize already 2 or 3 groups?

Regarding Table3 nausea and vomiting this is stated as not significantly different. This may be because of a too small group size, the number of patients studied?

Did the authors notice differences in the mood, mood changes or satisfaction of these patients? As one group received an antidepressant drug for treatment of late postoperative /neuropathic pain?

Regarding the differences in side effects. Could the authors tell something about differences in medication adherence?

The discussion can be improved by increased focus on the results in relation to the literature and thereby reduced by 25-30% in length?

L289 risk of addiction? How was this risk estimated in you study and how was the follow-up. Were there any signs of addiction? And were these observations comparable with the anticipated risk and the incidence comparable with what we know?

What does Hx mean in flow diagram?

In the figures of ROM, PAIN, Stiffness, and Function, is it possible to show SD, IQR or some variation on the 2 lines. Please remove *<0.05, when not applicable in the figure. However the changes in time may be significant!

Reviewer #3: Kim and co-authors present a reprospective review comparing efficacy of duloxetine and the combination drug, oxycodone/naloxone, for treatment of pain following total knee arthroplasty. This clinical review addresses and interesting question. While the overall study design is reasonable, the limitations of the approach/data - several of which were discussed by the authors - makes interpretation challenging undermining translation to clinical practice.

Concerns:

- Study design: the investigators correctly designate the evidence as Level IV as this is a retrospective chart review. The fact that each treatment being compared was provided by a single, distinct surgeon and assessments made in a non-blinded fashion raises concerns about the validity of the findings.

- Graphical presentation of data: while there are no reported difference between treatment groups it would still be appropriate to present error in some form (error bars displaying standard error of the mean is quite common).

- Language: there are numerous errors of syntax and grammar that make the manuscrip unfit for publication in its present form.

Other comments:

- The statement "Duloxetine has a similar effect to opioid on postoperative pain control" (Line 41) is an overly broad statement and would require removal or revision.

- Regarding the statement "Continued use of opioid increases the risk of opioid-related side effect, opioid induced hyperalgesia, and addiction" (Lines 54-56), the references sited do not suppor the contention that there is increased risk of opioid induced hyperalgesia.

- The statement "Although side effects were more frequent in the opioid group than in the duloxetine group, there was no difference in the incidence of side effects between the two groups (all p>0.05)" (Lines 213-214) is inappropriate. If there was no statistical difference, then the only appropriate statement would be that there is no difference between groups.

- The first sentence of the conclusion is overly broad and not supported by the data presented ("In conclusion, duloxetine can provide comparable pain management to opioid with improved function for up to one year after TKA without increasing apparent risks of adverse effects" (Lines 311-312)).

6. PLOS authors have the option to publish the peer review history of their article (what does this mean?). If published, this will include your full peer review and any attached files.

Reviewer #1: No

Reviewer #2: No

Reviewer #3: No

---

## [Author Response · Author response to Decision Letter 0]

25 Apr 2021

Response to reviewers

Reviewer #1: Yong In et al. present in their manuscript “Efficacy of Duloxetine Compared with Opioid for Postoperative Pain Control Following Total Knee Arthroplasty“ the alternative use of Duloxetine in contrast to opioid treatment in patients after total knee arthroplasty. This retrospective study demonstrate no significant difference in pain control using Duloxetine instead of opioid treatment over 6 weeks after arthroplasty in patients with VAS pain score >4 after one week of operation.

▶ We thank the reviewer for his/her valuable time and succinct summary of our study.

The authors describe, that the Duloxetine medication was prescribed with a dose of 30mg. This dose is a very low dose, as for i.e. neuropathic pain the recommended dose is 60mg daily. Why the dose had been set to 30mg and not to 60mg? This aspect should be clarified and be considered in the discussion section of the manuscript.

▶Thank you for your comments. In general, the dosage of duloxetine was started at 30 mg initially and then increased to 60 mg because duloxetine might have side effects [1]. In a previous randomized controlled trial, postoperative pain and quality of life of patients who underwent TKA for OA showed improvement with 30 mg of duloxetine [2]. In addition, 30 mg of duloxetine showed sufficient effect on fibromyalgia [3]. Therefore, in this study, duloxetine was used at 30 mg without increasing its dose later. We agree with the reviewer that a further study using different dosages of duloxetine for postoperative pain control is warranted. We have added this as a limitation of this study in the revised manuscript. (Lines 222-226, 313-315)

Furthermore Duloxetine has a distinct profile of side affects which should be kept in mind. In the manuscript this distinct side affects should be mentioned. However, this data are also mainly missing in the result section. 62.7% of the patients had a hypertension and 17.8% had a diabetes in the medical record. The question is in how many patients with diabetes had to treated additionally and in how many patients hypertension got worse.

▶Thank you for your comments. Duloxetine can increase blood pressure after administration due to its cardiovascular side effects as well as noradrenergic effect [4,5]. However, there is no restriction on the use of duloxetine in patients with high blood pressure as discontinuation alone can improve hypertension caused by the use of duloxetine [4,5]. In this study, blood pressure was measured at every outpatient follow-up after taking the drug. In one case, blood pressure was increased temporarily. However, no case that needed additional treatment, showed treatment changes, or had to stop taking the drug due to an increase in blood pressure. Cases requiring emergency treatment due to hypertension caused by taking duloxetine are less frequently reported. There are only a few case reports about them [4,5]. However, duloxetine can increase blood pressure. We agree with the reviewer that it is necessary to monitor it carefully. We have added this issue into the revised manuscript. (Lines 147-148, 214-215, 257-264)

Duloxetine resulted in modest increases in fasting plasma glucose levels in short- and long-term studies (0.50 and 0.67 mmol/l, respectively) [6]. However, the level of HbA1C was not increased in a placebo-controlled study [6]. In the short-term study, duloxetine 60 mg was used for 12 weeks. In the long-term study, it was used for up to 52 weeks [6]. In the present study, since 30 mg of duloxetine was used for 6 weeks, it was judged that its effect on glucose was less than its effects in studies described above. In the present study, glucose status was not regularly tested in diabetic patients. However, there were no specific findings in fating plasma glucose (FPG) at 2 weeks after drug administration. Hypertension patients and diabetes patients did not receive additional treatment or change treatment due to increased blood pressure or blood sugar after taking duloxetine [4-6]. We have added this issue into the method, results, and discussion sections of the revised manuscript (Lines 147-148, 214-215, 257-264). Of course, since the number of patients in this study was insufficient for evaluating its side effects, further studies with more patients are needed. We have added this issue as a limitation of this study in the revised manuscript. (Lines 313-315)

Looking at the risk of medical interaction Duloxetine is not recommended for the use in addition to metoprolol, further it can induce a Serotonin syndrome. Are there any data about patients suffering from a Serotonin Syndrome in this patient cohort?

▶Thank you for your comments. We fully agree with you. When metoprolol and antidepressant are used together, serious complications such as severe bradycardia and atrioventricular block can occur [7]. However, the risk is higher with the use of antidepressant drugs such as fluoxetine and bupropion while the risk is relatively lower with the use of escitalopram, citalopram, and duloxetine [7]. Nevertheless, when used together, serotonin syndrome can occur due to drug interaction [8]. Although serotonin syndrome is very rare, it is a life threatening event [8]. Therefore, it is necessary to observe carefully at all times. In this study, preoperative drug identification was used to identify drugs the patient was taking. If patients were using metoprolol and diuretics, the use of duloxetine was restricted. Most patients with hypertension used a calcium channel blocker and an angiotensin receptor blocker. Two patients who used beta blocker metoprolol were excluded from the duloxetine group from the beginning. In this study, no patient took metoprolol and duloxetine together. No patients developed serotonin syndrome either (Lines 150-153, 265-272). Of course, since the number of patients was insufficient for evaluating side effects in this study, further studies with more patients are needed in the future. We have added this issue as a limitation of this study in the revised version of the manuscript (Lines 313-315).

Mainly Duloxetine is increasing the risk of bleeding, especially in combination with NSAR medication. Are there any data in this study cohort? 

▶Thank you for your comments. When duloxetine and NSAID are used together, the risk of bleeding can increase due to platelet dysfunction [9]. However, since duloxetine is frequently used for OA-related pain, for which NSAIDs are not effective. Thus, these two types of drugs are often used in combination [2]. In this study, epistaxis, hematochezia, hematocrit reduction, and minor bleeding-related symptoms such as bruising were not observed in relation to taking duloxetine. No major bleeding findings requiring medical intervention were identified either. One patient complained of abdominal pain after taking the drug. Gastroscopy revealed no gastrointestinal abnormalities including bleeding. Of course, since the number of patients in this study was insufficient for evaluating side effects, a prospective study with more patients is warranted in the future. We have added this issue as a limitation of this study in the revised manuscript (Lines 149-150, 215-216, 273-278) (Lines 313-315).

Further Hyponatriemia is a severe side affect in the treatment of patients with Duloxetine especially in patients with diuretic treatment. How was this risk managed in this patients.

▶Thank you for your comments. It is well known that the use of antidepressants can increase the risk of hyponatremia [10]. However, among antidepressants, SNRI duloxetine has a lower incidence rate of hyponatremia compared to citalopram [10]. In addition, most cases of hyponatremia after the use of antidepressants appear within two weeks of initiation [10]. In this study, electrolyte tests were performed at two weeks after drug administration. There was no case of hyponatremia. Since the risk of hyponatremia is greatly increased when duloxetine and diuretic are used at the same time, in this study, drugs taken by patients were identified through preoperative drug identification. The use of duloxetine was restricted if patients were using diuretics. We have added this information to the revised manuscript (Lines 148-149, 215-216, 293-300) (Lines 313-315).

The risk profile of Duloxetine should be discussed in comparison to the risk profile of opioid use in the indication of prolonged treatment after total knee arthroplasty. From my point of view there should be clear statement that the use of Duloxetine is still an off label use, which has to be kept in mind especially as the risk profile of these very good medication is not low.

▶Thank you for your comments. We fully agree with the reviewer's opinion. Although duloxetine has been approved for use in OA patients in Korea when NSAIDs are not working, studies on side effects of continued use in TKA patients are insufficient. Various duloxetine-related side effects suggested by the reviewer were described in the discussion section of the revised manuscript (Lines 147-153, 214-216, 257-300) (Lines 313-315).

References

1. Blikman T, Rienstra W, van Raaij TM, ten Hagen AJ, Dijkstra B, Zijlstra WP, et al. Duloxetine in OsteoArthritis (DOA) study: study protocol of a pragmatic open-label randomised controlled trial assessing the effect of preoperative pain treatment on postoperative outcome after total hip or knee arthroplasty. BMJ Open. 2016;6(3):e010343. Epub 2016/03/05. doi: 10.1136/bmjopen-2015-010343. PubMed PMID: 26932142; PubMed Central PMCID: PMCPMC4785324.

2. Koh IJ, Kim MS, Sohn S, Song KY, Choi NY, In Y. Duloxetine Reduces Pain and Improves Quality of Recovery Following Total Knee Arthroplasty in Centrally Sensitized Patients: A Prospective, Randomized Controlled Study. J Bone Joint Surg Am. 2019;101(1):64-73. Epub 2019/01/03. doi: 10.2106/jbjs.18.00347. PubMed PMID: 30601417.

3. Arnold LM, Zhang S, Pangallo BA. Efficacy and safety of duloxetine 30 mg/d in patients with fibromyalgia: a randomized, double-blind, placebo-controlled study. Clin J Pain. 2012;28(9):775-81. Epub 2012/09/14. doi: 10.1097/AJP.0b013e3182510295. PubMed PMID: 22971669.

4. Mermi O, Atmaca M. [Duloxetine-Induced Hypertension: A Case Report]. Turk Psikiyatri Derg. 2016;27(1):67-9. Epub 2016/07/03. PubMed PMID: 27369688.

5. Shukla R, Khasbage S, Garg MK, Singh S. Duloxetine-induced hypertensive urgency in type 2 diabetes mellitus with diabetic neuropathy. Indian J Pharmacol. 2020;52(3):213-5. Epub 2020/09/03. doi: 10.4103/ijp.IJP_370_19. PubMed PMID: 32874005; PubMed Central PMCID: PMCPMC7446681.

6. Hardy T, Sachson R, Shen S, Armbruster M, Boulton AJ. Does treatment with duloxetine for neuropathic pain impact glycemic control? Diabetes Care. 2007;30(1):21-6. Epub 2006/12/29. doi: 10.2337/dc06-0947. PubMed PMID: 17192327.

7. Molden E, Spigset O. Interactions between metoprolol and antidepressants. Tidsskrift for den Norske laegeforening: tidsskrift for praktisk medicin, ny raekke. 2011;131(18):1777-9.

8. Gaffney RR, Schreibman IR. Serotonin syndrome in a patient on trazodone and duloxetine who received fentanyl following a percutaneous liver biopsy. Case reports in gastroenterology. 2015;9(2):132-6.

9. Frakes EP, Risser RC, Ball TD, Hochberg MC, Wohlreich MM. Duloxetine added to oral nonsteroidal anti-inflammatory drugs for treatment of knee pain due to osteoarthritis: results of a randomized, double-blind, placebo-controlled trial. Curr Med Res Opin. 2011;27(12):2361-72. Epub 2011/10/25. doi: 10.1185/03007995.2011.633502. PubMed PMID: 22017192.

10. Leth-Møller KB, Hansen AH, Torstensson M, Andersen SE, Ødum L, Gislasson G, et al. Antidepressants and the risk of hyponatremia: a Danish register-based population study. BMJ Open. 2016;6(5):e011200. Epub 2016/05/20. doi: 10.1136/bmjopen-2016-011200. PubMed PMID: 27194321; PubMed Central PMCID: PMCPMC4874104.

Reviewer #2: Dear authors,

Here you receive my review concerning the manuscript entitled “Efficacy of Duloxetine compared with opioid for postoperative pain control following total knee arthroplasty” for PLOS one (PONE-D-21-04527). The authors from the orthopedic surgery department present the results from their retrospective non-randomized single center observational study. The study was approved by the local institutional review board with number provided.

▶We thank the reviewer for his/her valuable time and succinct summary of our study.

This study number cannot be found in the database. Please explain? Line 82. Patients provided written informed consent. When were patients identified as eligible for this study and how were they approached for providing their informed consent. The reason for this question is in fact and rather to explain “retrospectively”.

▶Thank you for your comments. We have additionally uploaded the certificate of IRB approval. It was incorrectly stated that informed consent was obtained. This study was a retrospective study, which was approved by the IRB. However, this was a chart review study. Thus, patient consent was not required. We have modified this issue in the revised manuscript (Lines 73-76).

Primary question of this study was to determine whether Duloxetine could substitute opioids in a multimodal pain relief protocol after discharge and identification and difference in side effects of the 2 analgesics. Inclusion and exclusion criteria are well described. Language is clear and easy understandable.

▶We thank the reviewer for his/her valuable time succinct summary of our study.

Numbers of patients described in the “patients and treatment data” and abstract differ, i.e.857 patients vs 944 or 657 ? and 260 vs 290? I mean 153+137 is the mentioned 290, whereas the 121+118 is 239and not the 260 mentioned in the abstract. Moreover in the flow chart 944 differs from 857 in abstract or 944 in the patient and treatment data. Please rewrite with correct numbers?

▶Thank you for your comments. It was true that there were 290 patients. However, 32 out of 153 in the opioid group and 19 out of 137 in the duloxetine group were excluded. Finally, a total of 239 patients were included in the analysis. The number of patients listed in the Abstract as 260 was misrepresented. We have changed the sentence in the revised manuscript (Line 27).

So, patients received during the first postoperative period fentanyl PCA. Is it possible to differentiate between the needs of PCA fentanyl use during these first days in order to recognize possible differences in pain between patients. This may possibly have predictive value in need for analgesics after discharge. Maybe when possible correct these data for BMI differences between patients?

▶Thank you for your comments. We fully agree with the reviewer's comments. The use of PCA with fentanyl was a factor affecting pain patterns after surgery. It could also affect the use of duloxetine and opioid [2,3]. However, PCA was included in the pain management protocol of our critical pathway. All patients were administered the same dose. The usage amount of PCA showed no difference between the opioid group and the duloxetine group either. PCA consumption was 52.6 ml in the opioid group and 55.3 ml in the duloxetine group without showing significant difference (p = 0.492). When the use of PCA was compared based on BMI 24 with reference to a previous study [2], there was no significant difference between the group with BMI > 24 and the group with BMI ≤ 24 for both opioid and duloxetine users (Lines 143, 172-175).

What was the definition for kidney disease? Did the presence of kidney disease not have an impact on prescription of either drug? How was drug safety for these patients guaranteed?

▶Thank you for your comments. Only cases diagnosed as having impairment in renal function at the department of nephrology were designated as those having a kidney disease [1]. Since duloxetine is highly bound to plasma proteins and its metabolites are excreted by kidneys, caution should be exercised in cases of impaired renal function [1]. However, in patients with mild to moderate renal impairment, it is not necessary to adjust the dose of duloxetine. For patients with end stage renal disease (ESRD) or severe renal impairment, duloxetine is generally not recommended [1]. We did not administer duloxetine to patients with ESRD or severe renal impairment. We consulted with a nephrologist to determine whether to administer the drug. Duloxetine was not administered to patients with severe renal impairment through a preoperative renal function test. Of course, since the number of patients was insufficient in this study, we could not judge whether it was safe in terms of side effects of medication. Further studies with more patients are needed. We have added this issue as a limitation of this study in the revised manuscript (Lines 287-292, 313-315).

The patients , i.e. 15 in opioid group and 6 patients in the duloxetine group who received additional analgesics after six weeks. Did these patients also suffer from increased pain in the first postoperative week and needed more analgesic then also? Or did they have a (more) complicated course? Or tool these patients other comedication that may have had an effect on distribution or clearance, e.g. possible CYP enzyme competitors?

▶Thank you for your comments. Duloxetine and opioid were both discontinued after 6 weeks of administration and clinical features of the patients were observed. However, additional drugs were prescribed only if the patient wanted to continue taking it after 6 weeks of administration. These patients were excluded from the analysis because they took the drug for more than 6 weeks. These patients were not analyzed for clinical features. They might be more likely to complain of persistent pain patterns. However, this did not mean that the drug effectiveness was reduced due to co-medication with other drugs. These patients were finally included neither in the opioid group nor the duloxetine group. In the opioid group, 15 patients (9.8%) took opioid for 6 weeks or longer, and 6 (4.4%) patients took duloxetine for 6 weeks or longer in the duloxetine group. There was no significant difference between the two groups (p = 0.111). In addition, none of the patients who took the drug for more than 6 weeks in both groups took the drug for more than 3 months (Lines 129-133, 164-167).

In the case of selective cox-2 inhibitor, metabolism is achieved in the liver by cytochrome P450 (CYP) enzymes. However, duloxetine inhibits the activity of CYP enzymes. Thus, drug-drug interaction can occur when two drugs are used in combination [4,5]. However, in this study, all patients were administered the selective cox-2 inhibitor, celecoxib. When duloxetine was used in combination with celecoxib, it was not necessary to extend the duration. Since only a few patients took the drug for more than 6 weeks, it did not appear that the duration of drug use was prolonged due to an effect on distribution or clearance by co-medication. We have added this into the revised manuscript (Lines 279-286).

Is it possible to recognize difference between patients regarding pain sensation/severity after the first week?. In other words is it possible to recognize already 2 or 3 groups?

▶Thank you for your comments. Of course, at one week after the surgery, since the patient is still complaining of pain, it might be difficult to clearly distinguish the severity of the patient's pain. However, in this study, a self-reported questionnaire survey was conducted for patients at one week after surgery to determine the pain severity. Pain severity varied from patient to patient. This study was conducted for patients with a pain VAS score of 4 or more. We have added this issue into the revised manuscript (Lines 86-89).

Regarding Table3 nausea and vomiting this is stated as not significantly different. This may be because of a too small group size, the number of patients studied?

▶Thank you for your comments. We fully agree with your opinion. Of course, since the number of patients in this study was insufficient, we could not judge whether the drug was safe or having side effects. Further studies with more patients are needed for side effect comparison. We have added this issue into the revised manuscript (Lines 313-315).

Did the authors notice differences in the mood, mood changes or satisfaction of these patients? As one group received an antidepressant drug for treatment of late postoperative /neuropathic pain?

▶Thank you for your comments. Since duloxetine is an antidepressant drug, it is important to evaluate the mood. Although it would be nice to evaluate it using the Hamilton depression scale, etc., we did not take this approach in our study. We have added this issue as a limitation in the revised manuscript (Lines 315-317).

Regarding the differences in side effects. Could the authors tell something about differences in medication adherence?

▶Thank you for your comments. Side effects of opioid and duloxetine were similar, including nausea, vomiting, constipation, dizziness, drowsiness, headache, and dry mouth. Two patients who took opioid had intolerable nausea/vomiting and one patient who took duloxetine complained of dizziness. These patients wanted to stop taking the drug and discontinued the drug. Except for these three patients, there was no case of stopping the drug due to its side effects. Duloxetine can increase blood pressure and glucose level in hypertension and diabetes patients [1, 4-10]. In this study, there was no case of stopping or changing the drug due to increased blood pressure or glucose level by taking duloxetine. In addition, duloxetine can induce hyponatremia. When it is combined with NSAIDs, the risk of bleeding may increase. However, none of the patients in this study showed such side effects [1, 4-10]. Both duloxetine and opioid showed side effects similar to those of other studies (Lines 147-153, 214-216, 257-300) (Lines 313-315).

The discussion can be improved by increased focus on the results in relation to the literature and thereby reduced by 25-30% in length?

▶Thank you for your comments. We kept the most important facts and shortened the manuscript as much as possible.

L289 risk of addiction? How was this risk estimated in you study and how was the follow-up. Were there any signs of addiction? And were these observations comparable with the anticipated risk and the incidence comparable with what we know?

▶Thank you for your comments. In the opioid group, 15 (9.8%) patients took opioid for 6 weeks or longer, and 6 (4.4%) patients took duloxetine for 6 weeks or longer in the duloxetine group. There was no significant difference between the two groups (p = 0.111). In addition, none of these patients who took the drug for more than 6 weeks in both groups took the drug for more than 3 months (Lines 164-167). In this study, there was actually no opioid addiction or opioid use disorder due to the overuse of opioids. Symptoms of opioid addiction generally include inability to control opioid use, uncontrollable cravings, drowsiness, changes in sleep habits, weight loss, frequent flu-like symptoms, decreased libido, lack of hygiene, changes in exercise habits, and so on [13]. Orthopedic surgery uses more opioids than other surgical areas [14]. There is an increasing research interest in overuse of opioids [15, 16]. Chronic opioid use has dependence and risk of addiction. In fact, opioid addiction has been reported in 5-25% of chronic opioid use patients [17]. Therefore, the text expresses that there is a risk of addiction in the case of continuous use of opioids. However, no patients had used opioids for more than 3 months in this study. In addition, no patients showed addiction symptoms during the 1-year follow-up period. This paragraph has been deleted from the revised manuscript.

What does Hx mean in flow diagram?

▶Thank you for your comments. Hx means the history. We have changed “Hx” to “History” in the revised manuscript.

In the figures of ROM, PAIN, Stiffness, and Function, is it possible to show SD, IQR or some variation on the 2 lines. Please remove *<0.05, when not applicable in the figure. However the changes in time may be significant!

▶Thank you for your comments. We have added standard deviation (SD) and removed * < 0.05 from the revised manuscript. There was no difference between the two groups at each period. Both groups showed significant improvement in ROM, WOMAC pain, stiffness, function, and total score at 1 year postoperatively compared to the preoperative period. We have added SD into each figure of the revised manuscript (Figures 2 and 3).

References

1. Lobo ED, Heathman M, Kuan HY, Reddy S, O'Brien L, Gonzales C, et al. Effects of varying degrees of renal impairment on the pharmacokinetics of duloxetine: analysis of a single-dose phase I study and pooled steady-state data from phase II/III trials. Clin Pharmacokinet. 2010;49(5):311-21. Epub 2010/04/14. doi: 10.2165/11319330-000000000-00000. PubMed PMID: 20384393.

2. Chang KY, Tsou MY, Chan KH, Sung CS, Chang WK. Factors affecting patient-controlled analgesia requirements. J Formos Med Assoc. 2006;105(11):918-25. Epub 2006/11/14. doi: 10.1016/s0929-6646(09)60177-7. PubMed PMID: 17098693.

3. Viscusi ER, Ding L, Phipps JB, Itri LM, Schauer PR. High Body Mass Index and Use of Fentanyl Iontophoretic Transdermal System in Postoperative Pain Management: Results of a Pooled Analysis of Six Phase 3/3B Trials. Pain Ther. 2017;6(1):29-43. Epub 2016/12/23. doi: 10.1007/s40122-016-0064-z. PubMed PMID: 28004310; PubMed Central PMCID: PMCPMC5447541.

4. Garnett WR. Clinical implications of drug interactions with coxibs. Pharmacotherapy. 2001;21(10):1223-32. Epub 2001/10/17. doi: 10.1592/phco.21.15.1223.33891. PubMed PMID: 11601668.

5. Patroneva A, Connolly SM, Fatato P, Pedersen R, Jiang Q, Paul J, et al. An assessment of drug-drug interactions: the effect of desvenlafaxine and duloxetine on the pharmacokinetics of the CYP2D6 probe desipramine in healthy subjects. Drug Metab Dispos. 2008;36(12):2484-91. Epub 2008/09/24. doi: 10.1124/dmd.108.021527. PubMed PMID: 18809731.

6. Frakes EP, Risser RC, Ball TD, Hochberg MC, Wohlreich MM. Duloxetine added to oral nonsteroidal anti-inflammatory drugs for treatment of knee pain due to osteoarthritis: results of a randomized, double-blind, placebo-controlled trial. Curr Med Res Opin. 2011;27(12):2361-72. Epub 2011/10/25. doi: 10.1185/03007995.2011.633502. PubMed PMID: 22017192.

7. Gaffney RR, Schreibman IR. Serotonin syndrome in a patient on trazodone and duloxetine who received fentanyl following a percutaneous liver biopsy. Case reports in gastroenterology. 2015;9(2):132-6.

8. Hardy T, Sachson R, Shen S, Armbruster M, Boulton AJ. Does treatment with duloxetine for neuropathic pain impact glycemic control? Diabetes Care. 2007;30(1):21-6. Epub 2006/12/29. doi: 10.2337/dc06-0947. PubMed PMID: 17192327.

9. Leth-Møller KB, Hansen AH, Torstensson M, Andersen SE, Ødum L, Gislasson G, et al. Antidepressants and the risk of hyponatremia: a Danish register-based population study. BMJ Open. 2016;6(5):e011200. Epub 2016/05/20. doi: 10.1136/bmjopen-2016-011200. PubMed PMID: 27194321; PubMed Central PMCID: PMCPMC4874104.

10. Mermi O, Atmaca M. [Duloxetine-Induced Hypertension: A Case Report]. Turk Psikiyatri Derg. 2016;27(1):67-9. Epub 2016/07/03. PubMed PMID: 27369688.

11. Molden E, Spigset O. Interactions between metoprolol and antidepressants. Tidsskrift for den Norske laegeforening: tidsskrift for praktisk medicin, ny raekke. 2011;131(18):1777-9.

12. Shukla R, Khasbage S, Garg MK, Singh S. Duloxetine-induced hypertensive urgency in type 2 diabetes mellitus with diabetic neuropathy. Indian J Pharmacol. 2020;52(3):213-5. Epub 2020/09/03. doi: 10.4103/ijp.IJP_370_19. PubMed PMID: 32874005; PubMed Central PMCID: PMCPMC7446681.

13. Klimas J, Gorfinkel L, Fairbairn N, Amato L, Ahamad K, Nolan S, et al. Strategies to identify patient risks of prescription opioid addiction when initiating opioids for pain: a systematic review. JAMA network open. 2019;2(5):e193365-e.

14. Volkow ND, McLellan TA, Cotto JH, Karithanom M, Weiss SR. Characteristics of opioid prescriptions in 2009. Jama. 2011;305(13):1299-301. Epub 2011/04/07. doi: 10.1001/jama.2011.401. PubMed PMID: 21467282; PubMed Central PMCID: PMCPMC3187622.

15. Paulozzi LJ, Mack KA, Hockenberry JM. Vital signs: variation among States in prescribing of opioid pain relievers and benzodiazepines - United States, 2012. MMWR Morb Mortal Wkly Rep. 2014;63(26):563-8. Epub 2014/07/06. PubMed PMID: 24990489; PubMed Central PMCID: PMCPMC4584903.

16. Rudd RA, Seth P, David F, Scholl L. Increases in Drug and Opioid-Involved Overdose Deaths - United States, 2010-2015. MMWR Morb Mortal Wkly Rep. 2016;65(50-51):1445-52. Epub 2016/12/30. doi: 10.15585/mmwr.mm655051e1. PubMed PMID: 28033313.

17. Kahan M, Srivastava A, Wilson L, Gourlay D, Midmer D. Misuse of and dependence on opioids: study of chronic pain patients. Can Fam Physician. 2006;52(9):1081-7. Epub 2007/02/07. PubMed PMID: 17279218; PubMed Central PMCID: PMCPMC1783735.

Reviewer #3: Kim and co-authors present a reprospective review comparing efficacy of duloxetine and the combination drug, oxycodone/naloxone, for treatment of pain following total knee arthroplasty. This clinical review addresses and interesting question. While the overall study design is reasonable, the limitations of the approach/data - several of which were discussed by the authors - makes interpretation challenging undermining translation to clinical practice.

▶We thank the reviewer for his/her valuable time and succinct summary of our study.

Concerns:

- Study design: the investigators correctly designate the evidence as Level IV as this is a retrospective chart review. The fact that each treatment being compared was provided by a single, distinct surgeon and assessments made in a non-blinded fashion raises concerns about the validity of the findings.

▶Thank you for your comments. We totally agree with the reviewer's opinion. This study is a retrospective study with limitations in its design. Further research needs to be conducted through randomized controlled trial in the future. We have added this issue into the revised manuscript (Lines 79-81, 303-309).

- Graphical presentation of data: while there are no reported difference between treatment groups it would still be appropriate to present error in some form (error bars displaying standard error of the mean is quite common).

▶Thank you for your comments. We have added standard deviation into each figure of the revised manuscript (Figures 2 and 3).

- Language: there are numerous errors of syntax and grammar that make the manuscript unfit for publication in its present form.

▶We thank the reviewer for this insightful comment and we agree with the reviewer. We have asked an English editing service to improve our language for the revised manuscript.

Other comments:

- The statement "Duloxetine has a similar effect to opioid on postoperative pain control" (Line 41) is an overly broad statement and would require removal or revision.

▶Thank you for your comments. We have modified the sentence in the revised manuscript. (Lines 40-41)

- Regarding the statement "Continued use of opioid increases the risk of opioid-related side effect, opioid induced hyperalgesia, and addiction" (Lines 54-56), the references sited do not support the contention that there is increased risk of opioid induced hyperalgesia.

▶Thank you for your comments. We have added new reference in the revised manuscript.1 (Lines 53-54)

- The statement "Although side effects were more frequent in the opioid group than in the duloxetine group, there was no difference in the incidence of side effects between the two groups (all p>0.05)" (Lines 213-214) is inappropriate. If there was no statistical difference, then the only appropriate statement would be that there is no difference between groups.

▶Thank you for your comments. We have changed the sentence to “there was no difference in the incidence of side effects between the two groups (all p > 0.05)” (Line 206).

- The first sentence of the conclusion is overly broad and not supported by the data presented ("In conclusion, duloxetine can provide comparable pain management to opioid with improved function for up to one year after TKA without increasing apparent risks of adverse effects" (Lines 311-312)).

▶Thank you for your comments. We have removed the sentence from the revised manuscript. In conclusion, only facts based on results are presented in the revised manuscript (Lines 324-325).

Reference

1. Lee M, Silverman SM, Hansen H, Patel VB, Manchikanti L. A comprehensive review of opioid-induced hyperalgesia. Pain Physician. 2011;14(2):145-61. Epub 2011/03/18. PubMed PMID: 21412369.

---

## [Decision Letter · Decision Letter 1]

20 May 2021

PONE-D-21-04527R1

Efficacy of Duloxetine Compared with Opioid for Postoperative Pain Control Following Total Knee Arthroplasty

PLOS ONE

Dear Dr. In,

Thank you for submitting your manuscript to PLOS ONE. After careful consideration, we feel that it has merit but does not fully meet PLOS ONE’s publication criteria as it currently stands. Therefore, we invite you to submit a revised version of the manuscript that addresses the points raised during the review process.

We look forward to receiving your revised manuscript.

Kind regards,

Vijayaprakash Suppiah, PhD

Academic Editor

PLOS ONE

Journal Requirements:

Reviewers' comments:

Reviewer's Responses to Questions

**Comments to the Author**

1. If the authors have adequately addressed your comments raised in a previous round of review and you feel that this manuscript is now acceptable for publication, you may indicate that here to bypass the “Comments to the Author” section, enter your conflict of interest statement in the “Confidential to Editor” section, and submit your "Accept" recommendation.

Reviewer #1: All comments have been addressed

Reviewer #2: (No Response)

2. Is the manuscript technically sound, and do the data support the conclusions?

Reviewer #1: Yes

Reviewer #2: Yes

3. Has the statistical analysis been performed appropriately and rigorously? 

Reviewer #1: Yes

Reviewer #2: Yes

4. Have the authors made all data underlying the findings in their manuscript fully available?

Reviewer #1: Yes

Reviewer #2: Yes

5. Is the manuscript presented in an intelligible fashion and written in standard English?

Reviewer #1: Yes

Reviewer #2: Yes

6. Review Comments to the Author

Reviewer #1: (No Response)

Reviewer #2: Dear authors,

Here you receive my review concerning the manuscript entitled “Efficacy of Duloxetine compared with opioid for postoperative pain control following total knee arthroplasty” for PLOS one (PONE-D-21-04527R1). The authors from the orthopedic surgery department present the results from their retrospective non-randomized single center observational study for the second time in their revised manuscript.

Overall the authors took our previous comment very seriously. This improved the general understanding of the presented study, results and conclusion.

However, I have a few remarks.

Ad Abstract:

Line 39 please add period at the end of the sentence.

Lines 42-43 …considered an alternative to opioid… . Sure, I may understand this conclusion or suggestion. But please consider patients allergic or intolerant for morphine or opioids. As you show in this small group that there were no significant benefits regarding , i.e., nausea and vomiting.

Line 53 please change indiscreet into indiscrete

Line 55 please change post-operative into postoperative

Line 58 Here the abbreviation “OA” appears for the first time. Please write for the first time completely. You may mean osteoarthrosis or osteoarthritis?

Patients and treatment data line 104: here the authors mention the number of 944 eligible patients, whereas in the abstract (Methods)the total number mentioned initially is 857 patients. Please explain, rewrite, or adapt?

Line 102:As mentioned before in our previous review duloxetine 30 mg is considered a low dose. The authors answered that during the 6 weeks duloxetine dose was increased to 60 mg. This should be mentioned in Methods and/or results. For instance by a statement or observation that after X weeks X% of patients took a total of 60 mg of duloxetine daily.

Discussion Line326 please change the uppercase letter to lowercase regarding “Duloxetine”.

7. PLOS authors have the option to publish the peer review history of their article (what does this mean?). If published, this will include your full peer review and any attached files.

Reviewer #1: No

Reviewer #2: No

---

## [Author Response · Author response to Decision Letter 1]

21 May 2021

Dear authors,

Here you receive my review concerning the manuscript entitled “Efficacy of Duloxetine compared with opioid for postoperative pain control following total knee arthroplasty” for PLOS one (PONE-D-21-04527R1). The authors from the orthopedic surgery department present the results from their retrospective non-randomized single center observational study for the second time in their revised manuscript.

▶We thank the reviewer for his/her valuable time, and we agree with this succinct summary of our study.

Overall the authors took our previous comment very seriously. This improved the general understanding of the presented study, results and conclusion. However, I have a few remarks.

Ad Abstract:

Line 39 please add period at the end of the sentence.

▶Thank you for your comments. We added period at the end of the sentence in the revised manuscript. (Line 39)

Lines 42-43 …considered an alternative to opioid… . Sure, I may understand this conclusion or suggestion. But please consider patients allergic or intolerant for morphine or opioids. As you show in this small group that there were no significant benefits regarding , i.e., nausea and vomiting.

▶Thank you for your comments. We fully agree that there may be side effects from using opioid or duloxetine. More high quality large-scale randomized controlled trials are still required to further confirm the side effects of duloxetine. We added this issue in the abstract and conclusion of revised manuscript. (Lines 41-42 and 319-320)

Line 53 please change indiscreet into indiscrete

▶Thank you for your comments. We changed indiscreet into indiscrete in the revised manuscript. (Line 52)

Line 55 please change post-operative into postoperative

▶Thank you for your comments. We changed post-operative into postoperative in the revised manuscript. (Line 54)

Line 58 Here the abbreviation “OA” appears for the first time. Please write for the first time completely. You may mean osteoarthrosis or osteoarthritis?

▶Thank you for your comments. We added the full term of OA in the revised manuscript. (Line 57)

Patients and treatment data line 104: here the authors mention the number of 944 eligible patients, whereas in the abstract (Methods)the total number mentioned initially is 857 patients. Please explain, rewrite, or adapt?

▶Thank you for your comments. There was a mistake in abstract. The contents of Abstract have been modified in the revised manuscript. (Line 27)

Line 102: As mentioned before in our previous review duloxetine 30 mg is considered a low dose. The authors answered that during the 6 weeks duloxetine dose was increased to 60 mg. This should be mentioned in Methods and/or results. For instance by a statement or observation that after X weeks X% of patients took a total of 60 mg of duloxetine daily.

▶Thank you for your comments. In general, the dosage of duloxetine starts with 30 mg initially. It is increased to 60 mg, because this drug might have side effects. However, in our study, only duloxetine 30mg was used without dose increment, referring to previous studies that proved sufficient effect even with duloxetine 30mg. We added this issue in the method of the revised manuscript. (Lines 103-104)

Discussion Line326 please change the uppercase letter to lowercase regarding “Duloxetine”.

▶Thank you for your comments. We changed the uppercase letter to lowercase in the discussion of the revised manuscript. (Line 269)

---

## [Decision Letter · Decision Letter 2]

10 Jun 2021

Efficacy of Duloxetine Compared with Opioid for Postoperative Pain Control Following Total Knee Arthroplasty

PONE-D-21-04527R2

Dear Dr. In,

We’re pleased to inform you that your manuscript has been judged scientifically suitable for publication and will be formally accepted for publication once it meets all outstanding technical requirements.

Kind regards,

Vijayaprakash Suppiah, PhD

Academic Editor

PLOS ONE

Reviewers' comments:

Reviewer's Responses to Questions

**Comments to the Author**

1. If the authors have adequately addressed your comments raised in a previous round of review and you feel that this manuscript is now acceptable for publication, you may indicate that here to bypass the “Comments to the Author” section, enter your conflict of interest statement in the “Confidential to Editor” section, and submit your "Accept" recommendation.

Reviewer #2: All comments have been addressed

2. Is the manuscript technically sound, and do the data support the conclusions?

Reviewer #2: (No Response)

3. Has the statistical analysis been performed appropriately and rigorously? 

Reviewer #2: (No Response)

4. Have the authors made all data underlying the findings in their manuscript fully available?

Reviewer #2: (No Response)

5. Is the manuscript presented in an intelligible fashion and written in standard English?

Reviewer #2: (No Response)

6. Review Comments to the Author

Reviewer #2: (No Response)

7. PLOS authors have the option to publish the peer review history of their article (what does this mean?). If published, this will include your full peer review and any attached files.

Reviewer #2: No

---

## [Editor Report · Acceptance letter]

24 Jun 2021

PONE-D-21-04527R2 

Efficacy of Duloxetine Compared with Opioid for Postoperative Pain Control Following Total Knee Arthroplasty 

Dear Dr. In:

I'm pleased to inform you that your manuscript has been deemed suitable for publication in PLOS ONE. Congratulations! Your manuscript is now with our production department. 

Kind regards, 

on behalf of

Dr. Vijayaprakash Suppiah 

Academic Editor

PLOS ONE